# Designing highly multiplex PCR primer sets with Simulated Annealing Design using Dimer Likelihood Estimation (SADDLE)

Nina G. Xie 1,5, Michael X. Wang 1,5, Ping Song1, Shiqi Mao2, Yifan Wang3, Yuxia Yang3, Junfeng Luo3, Shengxiang Ren2✉ & David Yu Zhang4✉

One major challenge in the design of highly multiplexed PCR primer sets is the large number of potential primer dimer species that grows quadratically with the number of primers to be designed. Simultaneously, there are exponentially many choices for multiplex primer sequence selection, resulting in systematic evaluation approaches being computationally intractable. Here, we present and experimentally validate Simulated Annealing Design using Dimer Likelihood Estimation (SADDLE), a stochastic algorithm for design of multiplex PCR primer sets that minimize primer dimer formation. In a 96-plex PCR primer set (192 primers), the fraction of primer dimers decreases from 90.7% in a naively designed primer set to 4.9% in our optimized primer set. Even when scaling to 384-plex (768 primers), the optimized primer set maintains low dimer fraction. In addition to NGS, SADDLE-designed primer sets can also be used in qPCR settings to allow highly multiplexed detection of gene fusions in cDNA, with a single-tube assay comprising 60 primers detecting 56 distinct gene fusions recurrently observed in lung cancer.

¹ Department of Bioengineering, Rice University, Houston, TX, USA. ² Department of Medical Oncology, Shanghai Pulmonary Hospital, Tongji University School of Medicine, Shanghai, China. ³ NuProbe China, Shanghai, China. ⁴ NuProbe USA, Houston, TX, USA. ⁵These authors contributed equally: Nina G. Xie, Michael X. Wang. ✉email: harry_ren@126.com; genomic.dave@gmail.com

The advance of high throughput sequencing has uncovered a large number of biomedically relevant DNA sequences, from driver mutations in cancer to new bacterial/viral pathogen DNA sequences to microbiome metagenomic profiles that affect mental disorders on the gut-brain axis[1–4]. For discovery applications, "shotgun" whole genome sequencing (WGS) is the preferred approach to identify novel DNA sequences of interest[5]. However, the human genome comprises over 3 billion nucleotides, and despite the lowering costs of high-throughput sequencing, it is not practical today to perform WGS to high depths necessary for identification of subclonal mutations, such as somatic mutations in cancer. For routine detection of disease-relevant DNA variants in known genes of interest, targeted sequencing or direct qPCR approaches are typically used[6,7]. Of the two dominant methods today for target enrichment, multiplex PCR tends to have shorter workflows and require less DNA input than hybrid-capture probes[8]. However, multiplex PCR struggles to scale to large panels covering hundreds of genes, due to the nonlinear increase of primer dimer species that reduce NGS mapping rates and increase effective cost[9].

Currently, multiplex PCR methods for NGS target enrichment (e.g., Ampliseq[8]) primarily rely on (1) enzymatic digestion of modified bases in primers[10] and (2) DNA size selection to preferentially remove short amplicon species likely to be primer dimers. However, both steps are labor-intensive and cannot be applied universally to all multiplexed PCR reactions. In contrast, relatively little systematic work has been reported on computational approaches to minimizing the formation of primer dimers in the first place. To the best of our knowledge, existing multiplex primer design algorithm never exceeded 70 primer pairs in one tube[11–14]. This is mainly due to the high computational cost when the number of primers increases[15]. The development of a robust multiplex primer set design algorithm that produces highly multiplexed primer sets with minimal primer dimer formation could allow further scaling of multiplex PCR target enrichment to even larger NGS panels when combined with enzymatic and size selection methods. Alternatively, it can simplify the workflow of moderate size NGS and qPCR assays by removing the need for strict contamination control from open-tube steps.

There are two primary challenges in designing highly multiplexed PCR primer sets: First, for an $N$-plex PCR primer set comprising $2N$ primers, there are $\binom{2N}{2}$ possible simple primer dimer interactions. For $N = 50$, this corresponds to $\binom{100}{2} = 4950$ times as many potential primer dimer bindings as for a single-plex PCR primer set. Second, there are typically $M > 10$ reasonable candidate choices for each primer when considering specific gene targets and amplicon length constraints, resulting in $M^{2N}$ possible $N$-plex primer sets. For $M = 20$ and $N = 50$, the number of possible primer sets is $20^{100} \approx 1.3 \times 10^{130}$, billions of times larger than the number of atoms in the universe. Thus, it is computationally intractable to evaluate all possible multiplex primer sets. Simultaneously, primer dimer formation emerges from the interactions of two or more primers in the primer set, so changing the sequence of a primer to mitigate one primer dimer interaction may result in the appearance of another more serious primer dimer. In the language of numerical optimization, multiplex primer design is high dimensional problem with a highly non-convex fitness landscape. Consequently, standard convex optimization algorithms (e.g., gradient descent) will not be effective.

Here, we present Simulated Annealing Design using Dimer Likelihood Estimation (SADDLE), an algorithmic framework for designing highly multiplex PCR primer sets. Within this framework, we present an example multiplex primer design algorithm, comprising an algorithm for primer candidate generation and a rapidly computable Loss function for estimating primer dimers.

Using the SADDLE, we designed and experimentally tested multiplex primer sets comprising 192 primers (96-plex) and 784 primers (384-plex), and show low primer dimer formation through NGS experiments. Building upon this success, we built a single-tube 60 primer qPCR and Sanger assay to detect and identify 56 gene fusions with clinical actionability for non-small cell lung cancer.

## Results

**Simulated Annealing Design using Dimer Loss Estimation (SADDLE).** There are six main steps in SADDLE, as illustrated in Fig. 1:

1. Generation of forward primer (fP) and reverse primer (rP) candidates for each gene target.
2. Selection of an initial primer set $S_0$ from the primer candidates.
3. Evaluation of the Loss function L($S$) on the initial primer set $S_0$.
4. Generate a temporary primer set $T$ based on set $S_g$ (primer set from generation $g$) by randomly changing 1 or more primers.
5. Evaluate L($T$), and set $S_{g+1}$ to either $S_g$ (no change) or $T$, depending probabilistically on the relative values of L($S_g$) and L($T$).
6. Repeat steps 4 to 5 until an acceptable primer set $S_{final}$ is constructed.

The above abstract framework provides a basis for many potential multiplex primer design algorithms, depending on the specific details of primer candidate generation, form of Loss function, temporary set $T$ generation, and the dynamic probability of setting $S_{g+1}$ to $T$. Below, we describe our specific implementation of SADDLE, based on our accumulated understanding of primer design principles and primer dimer formation mechanisms. Given the infinite possibilities for function forms and hyper-parameters, we did not systematically evaluate or optimize at the high-level. Lower-level parameters, such as standard free energy ($\Delta G°$) ranges for primers, were experimentally optimized and these are described below.

1. Primer candidate generation. We begin our implementation of primer candidate generation through the selection of one or more "pivot" nucleotides on human genomic DNA around which we design the forward and reverse primers (Fig. 2a). The pivot nucleotides are the ones that must be included in the amplicon insert, and for example could be the hotspot region of a gene that is frequently mutated. From the pivot nucleotides and a constraint on the maximum length of the amplicons (e.g., determined by the read length of NGS), we can systematically generate a series of different proto-primers with 3′ end just outside the pivot nucleotides. The proto-primers have a large range of different lengths and binding energies to their complementary sequences, and will next be trimmed at the 3′ end to generate the primer candidates (Fig. 2a).

From our past experiences and preliminary optimization experiments, primers that hybridize to their cognate templates with $\Delta G° \approx -11.5$ kcal/mol have the best tradeoff between amplification efficiency/uniformity and nonspecific hybridization. Shorter primers may not bind consistently with high efficiency to their templates, resulting in variability in amplification efficiency and non-uniformity of amplicon on-target reads in the NGS library. Longer primers have increased likelihood of binding to other loci in human genome, and can result in non-specific amplicons. Based on this $\Delta G°$ goal, we next systematically constructed primer candidates from the proto-primers by truncating nucleotides from the 3′ end until the primer candidate has $\Delta G°$ between $-10.5$ kcal/mol and $-12.5$ kcal/mol (Supplementary Section S9). Due to the granularity

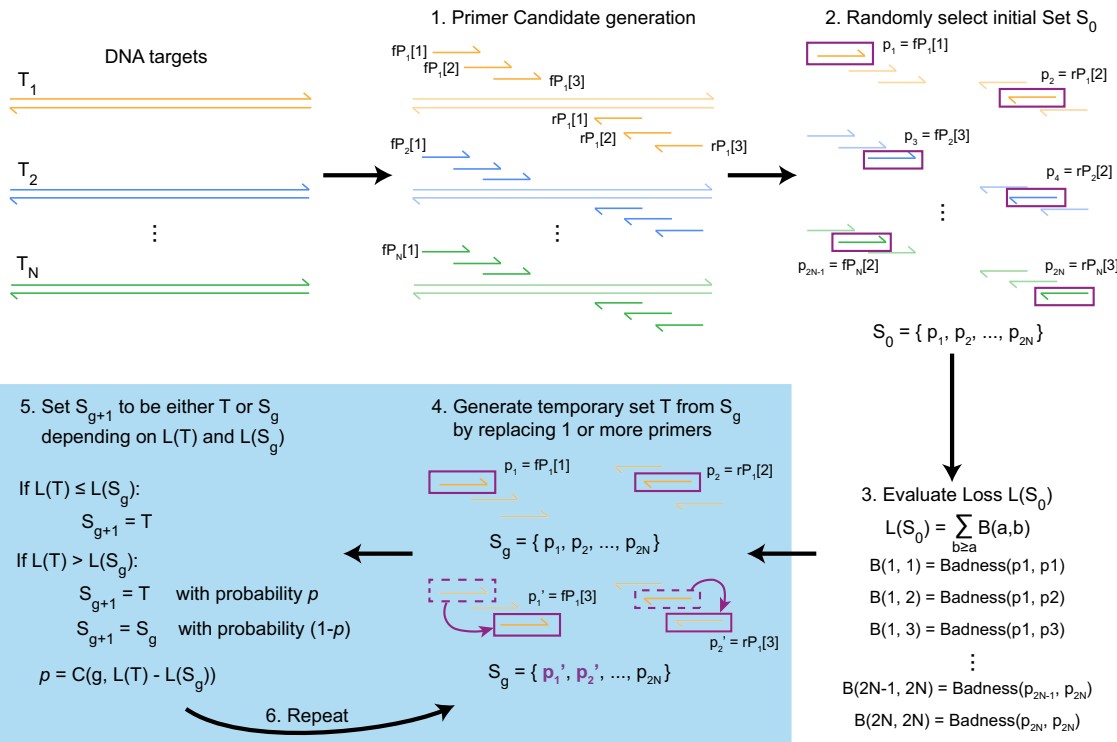

**Fig. 1 Overview of Simulated Annealing Design using Dimer Likelihood Estimation (SADDLE).** Given a set of DNA target sequences {$T_1$, $T_2$, ..., $T_N$}, the goal is to design a total of $2N$ PCR primers that can effectively amplify all DNA targets, while generating an acceptably low amount of primer dimer species. Steps 4 and 5 can be repeated a large number of times in order to improve (decrease) the Loss function value on the final primer set S. Multiple implementations, hyper-parameters, and parameters can be selected for each SADDLE step that can impact performance and speed.

of $\Delta G°$ for base stacks, some proto-primers with the same 5′ end will result in multiple primer candidates (e.g., with $\Delta G° = -10.9$ kcal/mol and $-12.0$ kcal/mol). Optionally, one could implement additional filters here to remove undesirable prime candidates, such as based on G/C content. For our demonstration panels, we restrict the G/C content of primer candidates to be between 0.25 and 0.75, removing primer candidates with G/C content outside this range.

In the implementation of SADDLE, primer candidates can be treated as individual primers or as primer pairs. Our specific implementation treats primers as pairs, so we next combinatorially generate all candidate primer pairs for an DNA target, in order to better constrain the distribution of amplicon lengths. Any candidate primer pairs that generate amplicons with length exceeding our maximum amplicon length or below our minimum are removed.

2. Initial primer set $S_0$ selection. We randomly selected a primer pair candidate for every amplicon that we wish to design, and collectively the selected primers are known as the initial primer set $S_0$.

3. Evaluation of Loss function $L(S)$ on $S_0$. The Loss function $L(S)$ is a rapidly computable function that aims to approximate the severity of primer dimer formation by a primer set $S$. $L(S)$ sums the potential primer dimer interactions between every pair of primers in the primer set. To prevent confusion, we refer to the predicted formation of dimers for a particular pair of primers to be the Badness. Mathematically,

$$L(S) = \sum_{b \geq a} \text{Badness}(p_a, p_b)$$
$$= \frac{1}{2} \cdot \sum_{a=1}^{2N} \sum_{b=1}^{2N} \text{Badness}(p_a, p_b) + \underbrace{\frac{1}{2} \cdot \sum_{a=1}^{2N} \text{Badness}(p_a, p_a)}_{\text{pre-calculated}} \quad (1)$$

where $p_a$ and $p_b$ are the $a$th and $b$th primer in primer set $S$,

respectively (Fig. 1). Note that the second term of $L(S)$ can be calculated in advance during primer candidates generation. One can imagine the Badness function to be proportional to the amount of primer dimers formed by two primers. In an optimized primer set with a relatively low concentration of primer dimers compared to the concentration of on-target amplicons, the amount of primer dimers formed between primer $p_a$ and $p_b$ should not significantly impact the amount of dimers formed between $p_a$ and $p_c$, so the Loss function being defined as the sum of the component Badness functions is justified.

The Badness function in our implementation is defined as follows (Fig. 2b):

$$\text{Badness}(p_a, p_b) = \sum \frac{2^{\text{len}} \cdot 2^{\text{numGC}}}{(d_1 + 1)(d_2 + 1)} \quad (2)$$

The sum in the Badness definition is over all reverse complementary subsequences between primer $p_a$ and $p_b$ with at least 4 nt of continuous complementarity. len is the length of the subsequence, $d_1$ and $d_2$ are the distances of the subsequence to the 3′ ends of primer $p_a$ and $p_b$, respectively, and numGC is the number of G/C nucleotides in the subsequence. Our choice of 4 nt is based on our preliminary experimental studies in qPCR showing that up to 3 nt of complementarity at the 3′ ends of two primers will not result in significant primer dimers even in no-template control (NTC) reactions.

For each complementary subsequence, its length (len) and its number of GC nucleotides in complementary subsequence (numGC) contribute exponentially to Badness. Thus, the exponential components of Badness roughly reflect the partition function of the complementarity interaction, with G/C base pairs roughly twice as strong as A/T base pairs. We chose to use these simplistic parameters, rather literature base stacking thermodynamics parameters[16–18], because there is significant uncertainty in the

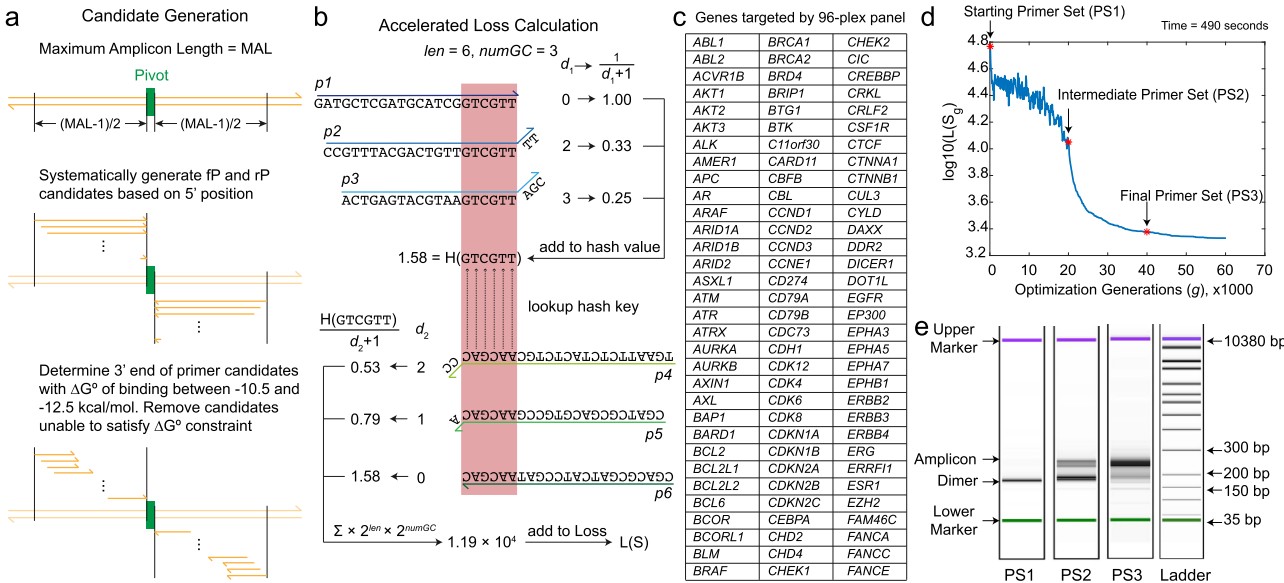

**Fig. 2 Implementation and experimental evaluation of a multiplex primer design algorithm based on the SADDLE framework. a** Method for generating candidate primer sequences for a DNA target T. **b** Implementation of Badness function that can be rapidly evaluated using hash tables. *len* is the length of the subsequence, *numGC* is the number of G/C nucleotides in the subsequence. $d_1$ and $d_2$ are the distances of the subsequence to the 3′ ends of primer $p_a$ and $p_b$, respectively. $p_1$, $p_2$, $p_3$ are examples of primer $p_a$, and $p_4$, $p_5$, $p_6$ are examples of primer $p_b$. **c** List of cancer genes selected as target sequences for a 96-plex primer set design. See Supplementary Excel spreadsheet for target selection details. **d** Loss function of primer sets S(g) across optimization generations g. The Loss function value decreases through the optimization and approaches a local minima after roughly 400 generations. We selected three different primer sets, constructed at generations 0, 200, and 400 for experimental evaluation; these are respectively called PS1, PS2, and PS3 for the remainder of this paper. Computation time for a 96-plex panel design is about 490 s for 60,000 iterations on a conventional laptop. **e** Capillary electrophoresis (Agilent Bioanalyzer 2100) analysis of amplicon products of PS1, PS2, and PS3. Here, 10 ng of the NA18562 human genomic DNA (Coriell) was used as input, and the median primer concentration was 45 nM in the PCR reaction. Seventeen cycles of PCR were performed using Vent (exo-) DNA polymerase (selected for its improved amplification ability for G/C-rich sequences). To facilitate more in-depth analysis by high-throughput sequencing (NGS), adapters/indexes were ligated to the amplicon products. No size selection was performed, in order to accurately reflect the fraction of primer dimer species following multiplex PCR. The on-target amplicons are expected to have an average length of roughly 250 nt.

effective salinity of the PCR reaction buffer, and because our previous studies on DNA thermodynamics suggests that previously reported $\Delta H°$ and $\Delta S°$ parameters do not extrapolate well to higher temperatures[19].

The distances of the complementary subsequence to the 3′ ends of primers $p_a$ and $p_b$, denoted as $d_1$ and $d_2$, are known to significantly affect the likelihood of primer dimer formation. In our preliminary qPCR experiments, we observed that a primer pair with 10 nt of complementarity at the 5′ end will not result in observable primer dimer formation, but a primer pair with 5 nt of complementarity at the 3′ end would. Depending on whether the specific DNA polymerase used, different $d_1$-based and $d_2$-based attenuation of Badness may be optimal for minimizing primer dimers. Because high-fidelity DNA polymerases with 3′>5′ exonuclease activity can remove mismatched 3′ nucleotides, the optimal $d_1$-based and $d_2$-based attenuation should be significantly weaker for high-fidelity DNA polymerases.

The evaluation of the Badness function is single largest component of software runtime cost, due to the large number of times the Badness function will be evaluated. For a primer that is 25 nt in length, there are 22 subsequences of length 4, 21 subsequences of length 5, etc. Evaluation of Badness for a single primer pair would thus have time complexity of $O(P^3)$, where $P$ is the length of each primer (Eq. 2). In our specific implementation, subsequence length *len* also has a maximum of 8 nt, decreasing the time complexity to $O(P^2)$. Naively, evaluation of the Loss L(S) of the whole primer set would have time complexity of $O(N^2 \times P^2)$ (Eq. 1). However, due to the additive nature of subsequence components to the overall Badness function, we implement more rapid Badness evaluation by using a hash table[20], as shown in Eq. (3), where H is

the hash table, s is a subsequence of the primer set, d is the distance to 3′ ends of each occurrence of s, and revcomp is a function that converts s to its reverse complement sequence (Fig. 2b). The time complexity to set up the hash table is $O(N \times P)$ to calculate the hash value for each unique subsequence in the primer set, and the time complexity to evaluate the L(S) by stepping through all subsequences of all primers is also $O(N \times P)$ (Eq. 3). Consequently, the overall time complexity of evaluating L(S) is $O(N \times P)$ for all primers in S.

$$H[s] = \sum \frac{1}{d+1}$$
$$\sum_{a=1}^{2N} \sum_{b=1}^{2N} \text{Badness}(p_a, p_b) = \sum \frac{2^{\text{len}} \cdot 2^{\text{numGC}}}{d+1} H[\text{revcomp}(s)] \quad (3)$$

4. Generate temporary primer set T based on $S_g$. Step 4 begins the recursive optimization process. Based on the current primer set $S_g$ at generation g, we first randomly select one primer pair to "mutate." For that primer pair, we randomly select a different primer pair from the list of all candidate primer pairs generated in Step 1. Temporary primer set T is thus generated by combining this new primer pair with all remaining primers in set $S_g$. Optionally, multiple primer pairs can be replaced simultaneously in this step to allow faster and more efficient exploration of the primer set space. In our preliminary in silico evaluations, we found that simultaneously mutating multiple primer pairs generally caused a slowdown of the optimization process.

5. Evaluate L(T) and set $S_{g+1}$ to be either T or $S_g$. The Loss of temporary primer set T can be evaluated significantly faster than

the initial evaluation of L($S_0$), because the hash table only need to be modified based on the changed primers.

We next compare the value of L($T$) vs. L($S_g$). If L($T$) is smaller than L($S_g$), then the primer pair change was an improvement and accepted, so $S_{g+1}$ is set to T. If L($T$) is larger than L($S_g$), the change was detrimental, but we will still accept the change with a certain probability, as part of the simulated annealing algorithm[21]. To clarify, "simulated annealing" here refers to a specific computer science algorithm, and not a literal simulation of a physical DNA thermal annealing process. If we never accept any detrimental primer pair changes, then the approach degenerates to become a stochastic gradient descent approach. In preliminary in silico evaluations, we confirmed that stochastic gradient descent produces final primer sets with significantly worse Loss, because it becomes too easy to get stuck in a local Loss minima.

The probability of accepting a detrimental change depends on both the magnitude of the detriment (L($T$) − L($S_g$)) and the generation $g$ of the optimization. Worse changes with higher L($T$) are accepted with lower probability, and later generations of the optimization (higher $g$) are less tolerant of detrimental changes. In our implementations, the probability of setting $S_{g+1}$ to be $T$ when L($T$) is greater than L($S_g$) are as follows:

$$p = \begin{cases} e^{\frac{L(S_g)-L(T)}{C(g)}} & (g < g_t) \\ 0 & (g \geq g_t) \end{cases} \quad (4)$$

where $e$ is Euler's number and $g_t$ is a positive interger. C($g$) is a function that is monotonically non-increasing in $g$, indicating decreasing tolerance to detrimental changes at later generations. The parameter $g_t$ indicates the generation in which simulated annealing terminates, and we switch over to stochastic gradient descent.

6. Repeat steps 4 and 5. Steps 4 and 5 are repeated until either a pre-determined generation $g$, or until L($S_g$) is below a pre-determined threshold $L_t$. In our implementation, we typically run the optimization to about $1.5 \times g_t$ to ensure we reach local minima. To further improve the overall quality of the generated primer set, we recommend running multiple SADDLE optimization processes with different starting conditions (initial primer sets) and selecting the best final primer set.

**Design and experimental evaluation of a 96-plex primer set.** We first used SADDLE to optimize the design of a 96-plex primer set, each amplicon target one arbitrarily selected exon of a different cancer-related gene[22–25] (Fig. 2c). Figure 2d shows the calculated value of L($S_g$) at different generations $g$, and is representative of our typical optimization trajectory. We selected the designed primer sets at three different optimization generations for experimental testing: PS1 (initial unoptimized primer set), PS2 (primer set with intermediate Loss optimization), and PS3 (primer set with saturating Loss optimization). The primer set Loss decreased roughly 24-fold from PS1 to PS3; after 40,000 generations, only very marginal improvements were observed. We chose the primer set at 40,000 generations as PS3, rather than the one at 60,000 generations, because we know that our Loss function is an imperfect predictor of primer dimers. Over-training on an imperfect Loss function can lead to worse experimental results. The optimization finished in about 10 min under a conventional laptop with MATLAB R2021b and Linux operating system.

We applied each of the three primer sets individually to human genomic DNA (10 ng NA18562, sheared to a mean length of approximately 150 nt) and amplified for 17 cycles. We next constructed NGS libraries from the amplicons generated using PS1, PS2, and PS3, using a standard adaptor ligation protocol (Supplementary Section S1). After library preparation,

capillary electrophoresis results show a clear increase of amplicons of the expected length from PS1 to PS2 to PS3 (Fig. 2e). In the NGS data analysis workflow, after the first step of adapter trimming, we separated NGS reads into three major species: on-target amplicons, dimers, and non-specific amplicons (Supplementary Section S2). On-target amplicons are the NGS reads that were successfully aligned to the intended amplicon sequences using Bowtie2[26]. The remaining NGS reads were aligned separately to each forward and reverse primer sequence. Reads with insert length shorter than the sum of the two aligned primers are classified as Dimers, and reads with insert length longer than the sum of the two aligned primers are classified as Non-specific amplicons (amplifying unintended regions of the human genome).

The amounts of these three species in the three primer set libraries are shown in Fig. 3a. Going from the PS1 to the PS3 library, the fraction of primer dimers dropped significantly, from 90.7% in the PS1 library to 39.6% in the PS2 library and then to 4.9% in the PS3 library. However, even with the decrease of dimers from the PS2 library to the PS3 library, the proportion of non-specific amplicons in these two libraries remained about the same. This is reasonable because the SADDLE Loss function was designed only minimizes primer Dimers, and does not consider likelihood of Non-specific amplicon formation. The distribution of amplicon length in NGS reads is consistent with the capillary electrophoresis results in three libraries (Fig. 3b and Supplementary section S3).

We next tested the PS3 primer set on five formalin-fixed, paraffin-embedded (FFPE) clinical tissue samples (one breast cancer, two lung cancer, and two colorectal cancer samples, see also Supplementary Section S5). The beeswarm plot of the observed reads (Fig. 3d) show high consistency across the different samples, and are also consistent with our results from sheared genomic DNA. The identities and quantities of primer dimers formed, likewise, are similar between FFPE DNA samples and genomic DNA (Fig. 3e).

To demonstrate the scalability of SADDLE, we next designed and tested a 384 amplicon panel comprising 768 primers. The optimization finished in about 60 min under a conventional laptop with MATLAB R2021b and Linux operating system. Due to the high cost of primer synthesis for this large panel, we only experimentally tested the final primer set design. Surprisingly, the observed Dimer fraction was only 1% for this library, using an input of 40 ng sheared NA18562 genomic DNA (Fig. 3f). Roughly 56% of the reads were Non-specific amplicons, resulting in a NGS library on-target rate of 43% (Supplementary section S6).

**Accuracy of the dimerization prediction.** We constructed the SADDLE Badness function based on our understanding of the mechanisms of primer dimer formation, but we know that this Badness function is imperfect both because our understanding of primer dimer formation is imperfect, and because it is computationally too expensive to implement many classes of potentially more accurate Badness functions. Accurate assessment of how good or bad the current Badness function is at predicting Dimers, however, is critical to further incremental improvement in multiplex PCR primer design using SADDLE.

Through the course of SADDLE optimization, we expect that the Dimer prediction accuracy will get worse in later optimization generations, because we are selecting for primer sets with low expected Badness that will include false negatives. Experiments and analysis of PS1, PS2, and PS3 confirm this understanding (Supplementary Section S7). The Dimer reads for each pair of primers from PS1 are plotted against the predicted Badness in Fig. 4a.

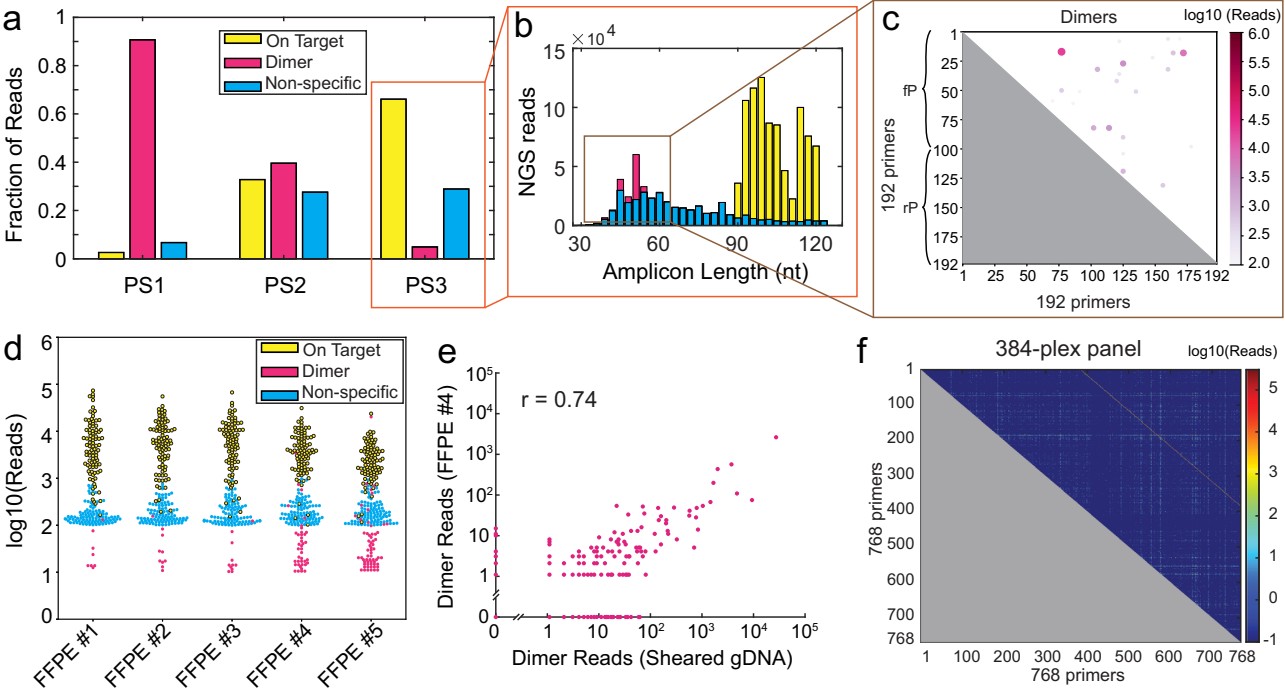

**Fig. 3 Experimental NGS results for SADDLE-designed primer sets. a** Distribution of reads observed in NGS library constructed using PS1, PS2, and PS3. On Target reads are defined as those that aligned to the intended amplicons; Dimer reads are defined as those whose insert lengths are smaller than the sum of the two primer lengths; all other reads were classified as Non-specific. The vast majority of Non-specific reads align to other regions of the human genome, via a non-cognate pair of forward and reverse primers. The fraction of NGS reads mapped to Dimers dramatically decreases from PS1 to PS2 to PS3. **b** Distribution of NGS reads in the three primer set libraries. **c** Distribution of observed primer dimers, based on aligned reads. Because forward primers (fP) can also form primer dimers with other forward primers, we aligned the first and last 25 nucleotides of each NGS read to the merged set of fP and rPs, with primers 1 through 96 in the diagram showing fPs and primers 97 through 192 showing rPs. For clarity of visualization, the log number of reads of observed primer dimers are displayed via both coloration and circle size. **d** Performance of the PS3 primer set of formalin-fixed paraffin-embedded (FFPE) tissue samples from deidentified lung cancer patients. Because the NGS libraries for these five samples differed slightly in total reads, here we plotted the distribution of reads normalized to 1 million reads. **e** The observed primer dimer species and their corresponding NGS reads were relatively similar between cell line genomic DNA and FFPE samples. **f** Demonstration of a 384-plex primer set designed by SADDLE (768 primers). The main diagonal shows On Target reads. Only about 1% of all reads were primer dimers (Supplementary Section S6).

To facilitate discussions of Badness function accuracy in terms of sensitivity and specificity, we set two separate thresholds: the Reads Threshold (horizontal orange line) and the Badness Threshold (vertical dotted purple line). The plotted Reads Threshold in Fig. 4a corresponds to the mean on-target read depth, and the Badness Threshold plotted correspond to the value that maximizes prediction sensitivity plus specificity. For these Threshold values, we observe a sensitivity 92.5% ($\frac{62}{67}$) and a specificity of 90.3% ($\frac{33,226}{36,797}$). By adjusting the Badness Threshold value, we can change the tradeoff between sensitivity and specificity, resulting in a receiver operator characteristic (ROC) curve (Fig. 4b). The area under the ROC curve (AUROC) is 0.9577, indicating very high Dimer prediction accuracy by the Badness function. When the Read Threshold is adjusted higher, the AUROC also increases (Fig. 4c), but the positive predictive value (PPV) decreases.

We next examined the top five most dominant Dimer reads in the library (Fig. 4d) and compared them to the top five predicted dimer reads based on the Badness function (Fig. 4e). It is noteworthy that only one of the two different top five lists overlap. The other four predicted dimers did not contribute significantly experimentally, and the other four observed dimers were not predicted to have high risk for dimer formation. At a glance, it appears we over-weighted the possibility of forming primer dimers in which the 3′-most nucleotide in unpaired, and we may need to adjust the Badness function to allow a stronger attenuation of Badness based on distance from the 3′ end. Additionally, it appears that the Badness function may be not scaled optimally, as the log10(Badness) ranges between 0 and 3.5, whereas the log10(Dimers) ranges between 0 and 5 (Fig. 4f). This may mean that the current algorithm over-weights weak potential dimers, at the expense of insufficiently avoiding strong predicted primer dimers.

Beyond the above observations, it is not clear why some dimers are observed at much higher reads experimentally than others. For example, the top observed dimer only has a 5 nt overlap at the 3′ end, compared to a 7 nt overlap at the 3′ end for the rank 4 dimer. This is not consistent with our understanding of DNA hybridization and polymerase extension kinetics, and implies that we may not be able to generate a perfect Badness function even ignoring computational resource constraints.

**Gene fusion detection with qPCR and sanger sequencing.** Gene fusions are therapeutic targets and attractive diagnostic bio-markers to guide treatment[27–30]. Currently, gene fusions are detected either in single-plex by qPCR for known high-frequency fusions (e.g., *BCR-ABL1*), or by NGS. A highly multiplexed qPCR assay that can detect tens of potential gene fusions relevant to a particular disease could greatly increase the accessibility of gene fusion testing.

Here, we used SADDLE to design a set of 60 primers to detect 56 actionable gene fusions for non-small cell lung cancer

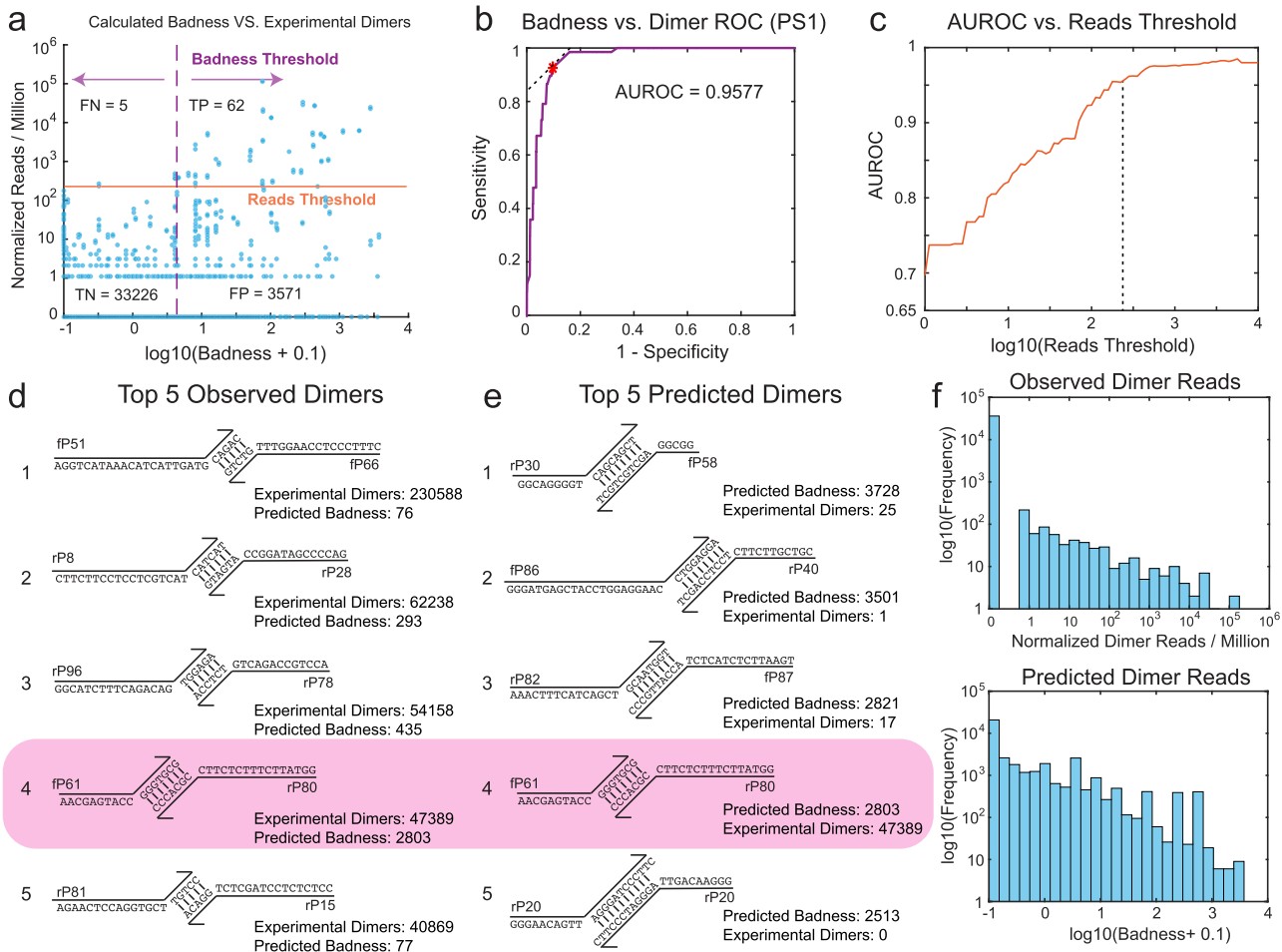

**Fig. 4 Evaluation of prediction accuracy of the Badness function for individual primer dimer candidates. a** Comparison of observed vs. predicted primer dimers for all possible pairs of primers in the PS1 library. The horizontal orange line shows the mean on-target reads for the 96 intended amplicons. By changing the Badness Threshold, different tradeoffs of dimer prediction sensitivity and specificity can be achieved. For the current Reads Threshold and Badness Threshold, we observe 92.5% sensitivity and 90.3% specificity. **b** Receiver operator characteristic (ROC) curve for dimer prediction sensitivity and specificity achieved by changing the Badness Threshold. **c** The Area Under the ROC (AUROC) value depends on the Reads Threshold, with a highest achievable AUROC of about 0.98. **d** The top five dimer species experimentally observed to form with highest number of aligned NGS reads. **e** The top five potential dimer species predicted to form based on the Badness function. Note that only the #4 species are present are both list in panels **d** and **e**. **f** Distribution of observed NGS reads and predicted Badness for all possible primer dimers.

(NSCLC) across six genes (*ALK, ROS1, RET, NRTK1, NTRK2,* and *NRTK3*). The number of primers are lower than $56 \times 2$ because the same exon can be fused with multiple partner genes or exons. We detect the fusions in complementary DNA (cDNA) reverse transcribed from RNA, in order to limit the complexity and length of the detection targets. For each fusion of interest, the primer set includes a forward primer (fP) targets the upstream partner gene and a reverse primer (rP) targets the downstream partner gene (Fig. 5a).

We first tested the multiplex PCR panel against synthetic samples bearing the gene fusions of interest (Fig. 5b, c). In all cases, the positive samples were clearly distinguishable by cycle threshold (Ct) value against both commercial wildtype cDNA (WT) and the no-template control (NTC), with all ΔCt values above 10. We also tested the panel on synthetic gene fusion samples with a variant allele frequency (VAF) of 1% (Supplementary Section S8). The 1% VAF samples were constructed by mixing synthetic gBlocks that contained a single fusion (the variant) with human cDNA (the wildtype).

Finally, we applied the gene fusion qPCR panel to clinical cDNA samples extracted from extracellular vesicles in blood plasma from NSCLC patients (Fig. 5d). Of the ten clinical samples analyzed,

three were called positive for gene fusions. To identify the exact gene fusion in these samples, we performed Sanger sequencing on the amplicons from the positive samples. Two samples were identified with *EML4* exon20-*ALK* exon20, and one was identified with *EML4* exon 15-*ALK* exon 20.

## Discussion

In this study, we designed a multiplex PCR primer design algorithm SADDLE targeting numerous genomic regions in a single tube. We presented experimental validation of primer sets on a 96-plex cancer-related exons panel, demonstrating that the SADDLE was capable in selecting better primers by reducing dimerization in a multiplex PCR reaction. The dimer rate decreased going from the 90.7% in a naively designed PS1 to 39.6% in an intermediate PS2 and to 4.9% in an optimized PS3, resulting in an increased on-target rate as well as greater uniformity of on-target amplicons. In another 384-plex panel targeting random-selected SNPs in the human genome, the NGS library using the optimized primer set showed a dimer rate of 1%. SADDLE can reduce reagent costs and enable the amplification of hundreds of target templates simultaneously without wasting NGS reads. Importantly, library

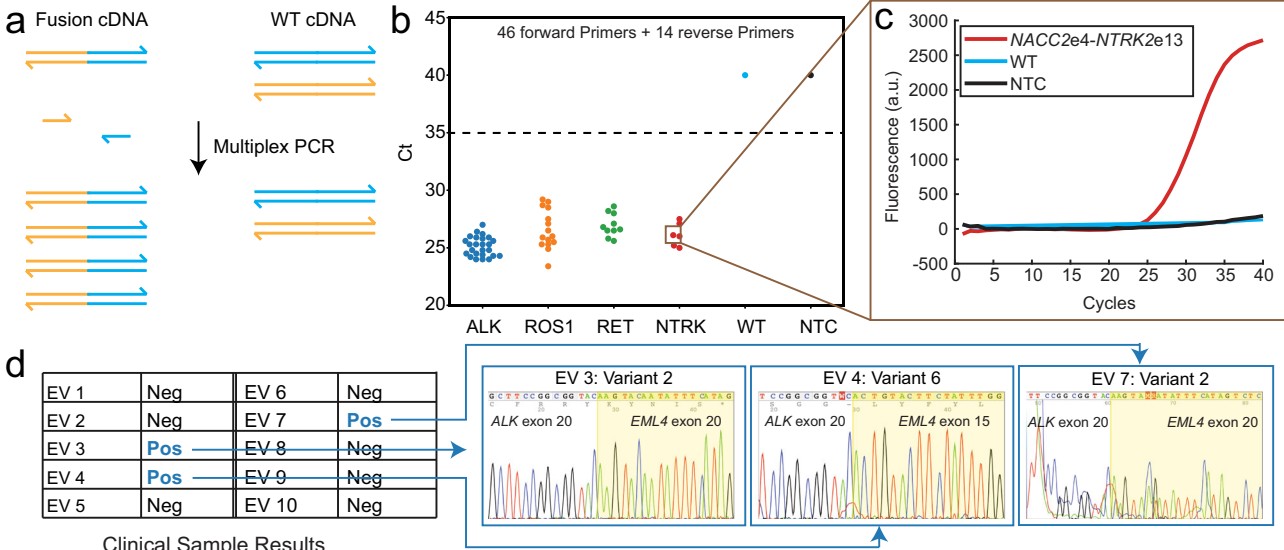

**Fig. 5 Highly multiplexed qPCR detection of gene fusions using SADDLE-designed primer sets. a** Complementary DNA (cDNA) prepared through reverse transcription of RNA can have known target sequences at the exon breakpoints. Although it is trivial to design a single-plex qPCR assay to detect a single known fusion, such as BCR-ABL1[36], we are not aware of any reports of highly multiplexed qPCR assays to simultaneously detect ≥10 different gene fusion cDNA species. For this assay, we designed a 60 primer set (46 forward, 14 reverse) that together can amplify 56 distinct gene fusion types commonly observed in non-small-cell lung cancer[37]. **b** Summary of observed qPCR cycle threshold (Ct) values for the 56 reactions, each with 1 of the 56 synthetic fusion DNA species across 6 genes (ALK, ROS1, RET, and NTRK1/NTRK2/NTRK3), each with 1700 copies. WT indicates wildtype commercial cDNA, and NTC indicates no template control. See Supplementary Section S8 for additional details and experimental results, including Sanger sequencing traces of each reaction product. **c** Example qPCR trace showing detection of the fusion DNA sequence joining NACC2 exon 4 to NTRK2 exon 13. **d** Clinical sample results on cDNA reverse transcribed from RNA from extracellular vesicles. Samples 3, 4, and 7 tested positive for a gene fusion, and sequence alignment of the Sanger sequencing results (right panels) show the exact identifies of the fusions.

preparation using optimized primer sets generated by SADDLE does not depend on labor-intensive enzymatic cleavage or size selection steps to remove dimers.

The improvement of NGS library on-target rates through the reduction of primer dimers can allow significantly larger targeted panels to be possible using multiplex PCR library preparation. Because multiplex PCR generally requires less input DNA and are faster than ligation-based library preparation approaches[31], due to the low yields of end repair and ligation, we envision that SADDLE-designed primers can be useful for a variety of research and clinical applications where DNA sample quantities are limited and/or where rapid turnaround is needed. For example, in oncology tissue biopsies obtained through fine needle aspirates and core biopsies are frequently insufficient for standard NGS analysis, and cell-free DNA from peripheral blood plasma likewise are limited and impose sensitivity limitations to ligation-based approaches[32]. Furthermore, in reproductive medicine, samples from amniocentesis and preimplantation genetic screening (PGS) and preimplantation genetic diagnosis (PGD) are also very limited, and require rapid turnaround for molecular diagnostics due to the time-sensitivity of clinical decisions[33].

Through our analyses of predicted vs. observed dimers, we found that the parameters in the Loss function used in SADDLE could be adjusted to optimize dimer prediction performance, particularly in the 3′ distance attenuation. However, with the current SADDLE algorithm, Non-specific amplicons now appear to dominate off-target rates, rather than Dimers. Thus, to further scale-up the panels that can be designed by SADDLE, it will be necessary to construct and optimize new Loss functions that penalize primer sets based on predicted off-target genomic amplification. Modification of the Loss function to minimize Non-specific amplicon formation would require significantly more work, as it requires consideration of the expected sample genome sequence. Whereas the current Loss function is

"universal" in improving multiplex PCR primer set designs, a Loss function that considers Non-specific amplicons would inherently be suboptimal for primer dimer minimization. A Loss function predicting Non-specific amplification must also consider external factors, including the average length of the DNA molecules in the sample and nonpathogenic genomic polymorphisms. Current Loss function can be further improved based on NGS data and other methods including machine learning[34].

In medical and research applications where the cost of NGS cannot be economically justified[35], qPCR assays will likely be the dominant tool for study of genomic variants. In qPCR, even single-plex primers can form significant dimers if poorly designed with Ct values below 30. Multiplex qPCR thus typically requires significant empirical optimization, even at around 4-plex[11]. SADDLE allowed us to successfully design a 60-primer qPCR panel targeting 56 gene fusions, and exact fusion identities can be determined through affordable Sanger sequencing. Thus, we envision that SADDLE can revolutionize the use of qPCR for highly multiplex molecular diagnostics.

## Methods

**Ethical approval.** All procedures performed in studies involving human participants were approved by the ethics committees of Shanghai Pulmonary Hospital, Tongji University (protocol K19-155Y), and were in accordance with the 1964 Helsinki declaration and its later amendments or comparable ethical standards. Informed consent was obtained from all participants.

**Oligonucleotides.** All primers were purchased as standard desalted DNA oligonucleotides (Integrated DNA Technologies), and stored at 4 °C.

**Samples.** Synthetic DNA templates were purchased as desalted DNA oligonucleotides (gBlocks, Integrated DNA Technologies), and stored at −20 °C. Human cell-line gDNA sample NA18562 (Coriell Biorepository) was stored at −20 °C. The gDNA was mixed with synthetic DNA templates at various ratios to create samples containing different proportions of a specific variant sequence. Dilution of gDNA

samples and synthetic DNA templates were made in 1× TE buffer with 0.1% Tween 20 (Sigma Aldrich).

FFPE slides were purchased from Coriell Institute. FFPE DNA was extracted from GeneRead DNA FFPE Kit (Qiagen).

Ten plasma samples from ten NSCLC patients in de-identified format were collected from Shanghai Pulmonary Hospital. RNA in extracellular vesicles was extracted with exoRNeasy Serum/Plasma Kit (Qiagen). cDNA was synthesized with SuperScript™ IV First-Strand Synthesis System (ThermoFisher Scientific).

**Multiplex PCR protocol**. Multiplex PCR was performed on a T100 Thermocycler or a C1000 Thermocycler (Bio-Rad). The total volume of each reaction was 50 µl. DNA sample input ranged from 10 to 100 ng per tube. PCR reagents including vent (exo-) polymerase, ThermoPol Reaction Buffer (10×), and dNTP (New England Biolabs) were used for enzymatic amplification. Thermal cycling started with a 3 min incubation step at 95 °C for polymerase activation, followed by 17 cycles of 30 s at 95 °C for DNA denaturing, 3 min at 60 °C for annealing, and 30 s at 72 °C for extension, followed by a final extension of 5 min at 72 °C. Detailed experiment protocol for the NGS library preparation can be found in Supplementary Section S1, S3.

**End repair protocol**. Multiplex PCR product was end-repaired using NEBNext® Ultra™ II End Repair/dA-Tailing Module (New England Biolabs). Each reaction was a mixture of 3 µl NEBNext Ultra II End Prep Enzyme Mix, 7 µl NEBNext Ultra II End Prep Reaction Buffer, 20 µl multiplex PCR products, and 30 µl H2O. End repair was performed on a Eppendorf Mastercycler. Thermal cycling started with the incubation at 20 °C for 30 min and 65 °C for 30 min, with the heated lid set to 80 °C.

**Adapter ligation**. End repair mixture was ligated with adapters using NEBNext® Ultra™ II Ligation Module (New England Biolabs). Each reaction was a mixture of 30 µl NEBNext Ultra II Ligation Master Mix, 1 µl NEBNext Ligation Enhancer, 2.5 µl NEBNext Adaptor for Illumina, and 60 µl previous End repair mixture. Ligation was performed on a Mastercycler from Eppendorf. Thermocycling started with the incubation at 20 °C for 15 min with the heated lid off; after adding 3 µl USER™ enzyme to the ligation mixture, the reaction was incubated at 37 °C for 15 min with the heated lid set to 55 °C.

**Index quantitative PCR**. Following adapter ligation, Index qPCR was performed on CFX96 Touch Deep Well Real-Time PCR Detection system (Bio-Rad). Quantification of different libraries was performed simultaneously in each well. Each reaction was a 10 µl mixture, with 1 µl i5 index, 1 µl i7 index, 1 µl ligation products, 2 µl Milli-Q, and 5 µl PowerUp SYBR Green Master Mix. Experiment was performed following a thermal cycling protocol with a 3 min incubation step at 95 °C for polymerase activation, followed by 40 cycles of 10 s at 95 °C for DNA denaturing and 30 s at 60 °C for annealing and extension. Ct values were obtained directly from the CFX96 system.

**Index PCR**. Index PCR was performed on a T100 Thermocycler or a C1000 Thermocycler (Bio-Rad). Index primers used were NEBNext® Multiplex Oligos for Illumina® (New England Biolabs). Each reaction was a mixture of 2 µl each i5 and i7 index primers, 5 µl ligation products, and PCR reagents including vent (exo-) polymerase, ThermoPol Reaction Buffer (10×), and dNTP. The volume of each reaction was 52 µl. Thermal cycling started with a 3 min incubation step at 95 °C for polymerase activation, followed by various cycles of 30 s at 95 °C for DNA denaturing and 30 s at 60 °C for annealing, and 30 s at 72 °C for extension, followed by a final extension of 5 min at 72°.

**Column purification**. Multiplex PCR products and ligation products were all purified using DNA Clean & Concentrator Kits (ZYMO Research). The volume of DNA-binding buffer was 250 µl for multiplex PCR products clean-up, and 482.5 µl for ligation products clean-up; 25 µl Milli-Q water was used as elution buffer for each reaction.

**Beads purification**. Index PCR product was purified using AMPure XP beads (Beckman Coulter). For each 50 µl reaction mixture, 90 µl of beads was added; 40 µl Milli-Q water was used as elution buffer.

**Library quantitation**. All the libraries were quantified using the Qubit™ dsDNA HS Assay Kit (ThermoFisher Scientific).

**Bioanalyzer**. Sizes of PCR products and libraries were measured using Bioanalyzer High Sensitivity DNA Assay (Agilent), and DNA chips were run on the Agilent 2100 Bioanalyzer system.

**Next-generation sequencing**. All the libraries were loaded on a Miseq Reagent V2 for obtaining pair-end reads and were sequenced on a Miseq (Illumina).

**Sanger sequencing**. PCR products were purified and prepared using a BigDye Terminator v1.1 Cycle Sequencing Kit (Thermo Fisher Scientific) and were sequenced on a Thermo Fisher Scientific 3500 Series Genetic Analyzer. Detailed experiment protocol for Sanger Sequencing can be found in Supplementary Section S8.

**Reporting summary**. Further information on research design is available in the Nature Research Reporting Summary linked to this article.

## Data availability
The reference and sample-specific gDNA sequence data are available from the NCBI Nucleotide database, the Ensembl database, the COSMIC database, and the Foundation Medicine gene list. Primer sequences for all experiments can be found in Supplementary Data 1. Badness of all primer pairs of PS1, PS2 and PS3 can be found in Supplementary Data 2. Number of dimer reads of all primer pairs of PS1, PS2, and PS3 can be found in Supplementary Data 3. Raw NGS data is available at https://doi.org/10.6084/m9.figshare.16944154.v3.

## Code availability
The MATLAB code used for multiplex PCR primer algorithm is available upon request under NDA for academic lab. The MATLAB code and Python code for NGS data processing are available at https://github.com/NinaGXie/SADDLE.

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

## Acknowledgements

The authors thank Paul Dolber, Lauren Yuxuan Cheng, Carol Kerou Zhang, and Gavin Jiaming Li for editorial assistance. This work was funded by Shanghai Shenkang Hospital Development Center grant SHDC12019133 to S.R. and NIH grant R01CA203964 to D.Y.Z.

## Author contributions

N.G.X., M.X.W., S.R., and D.Y.Z. conceived the project. N.G.X. performed the NGS experiments. N.G.X., M.X.W., and D.Y.Z. wrote the program for primer design and data analysis. N.G.X., M.X.W., and P.S. analyzed the NGS data. S.M., Y.W., Y.Y., J.L., and S.R. designed and performed the qPCR-based and Sanger-based gene fusion experiments. S.M. and Y.W. analyzed the qPCR and Sanger data. N.G.X., M.X.W., S.R., and D.Y.Z. wrote the manuscript with input from all authors.

## Competing interests

There is a patent pending on the Multiplex Primer Design Algorithm presented in this manuscript, N.G.X., M.X.W., and D.Y.Z. are the inventors, patent applicant is Rice University. This patent has been exclusively licensed to Nuprobe Global. N.G.X., M.X.W., and P.S. declare a competing interest in the form of consulting for Nuprobe USA. D.Y.Z. declares a competing interest in the form of consulting for and significant equity ownership in Nuprobe Global, Torus Biosystems, and Pana Bio. The remaining authors declare no competing interests.
