## [Peer Review File · Nature Communications]

Reviewers' Comments:

Reviewer #2:

Remarks to the Author:

The authors describe a computational implementation for multiplex PCR design they term SADDLE, and use this methodology to generate a series of multiplex panels for a variety of PCR applications (e.g., NGS sequencing, sanger, qPCR detection).

The methodology is described in computational and mathematical terms. However, the key application seems to centre on what the authors term their Badness function, which thematically appears to be based predominantly on 5mr basespair pattern matching for all primer pairs within a pool; followed by bootstrapping randomly sampled primers to maximize dangling 3' ends of the predicted interactions, which the authors believe minimizes dimer formation.

Overall the manuscript is very nicely laid out, and the authors have attempted to characterize their platform across a variety of relevant, clinical situations (e.g., FFPE and fusion events). The qPCR multiplex assay for fusion detection in particular I thought was a very nice clinical application for rapid detection of fusion events from the cDNA of cancer patients.

Methodologically, the authors have characterized their system quite well, and I commend them for the amount of detail as well as the attempt to make their method clinically relevant; however, I'm not entirely convinced of the novelty of the proposed work. Although the mathematical description and validation work is well described, my assessment is that the methodology can mostly be reduced to the pattern matching loop that iterates through all possible primer pairs, followed by weighting for nonbinding events at the 3' end of the primer, and neither of these solutions would be particularly new to the field.

Furthermore, the rather significant off-target amplification of their panels substantially limits the impact of their methodology. While the authors are transparent in the off target effects observed in their data, I would have to be somewhat critical of the comment made that their pipeline is not optimized to control for off target effects, particularly since their methodology would be most readily compared to the Ampliseq application, which I would generally characterize as the market leader in this capacity. To draw a comparison, Ampliseq's CCP cancer panel I can state from experience has multiplex pools with 8000 primers in single tube, and achieves an on target rate in the high 90% with virtually no primer dimer formation whatsoever. The Ampliseq transcriptome and exome panels are even higher (40,000 primer pairs and 200,000 respectively, if memory serves), and the presence and success of this commercial solution therefore takes away from the impact and relevance of the authors work rather substantially, unfortunately.

As such, while the authors have written a very nice paper and done a lovely job characterizing and applying their work, I'm not entirely convinced as to the novelty and overall relevance of their methodology to the field, particularly given these limitations and the other commercial solutions currently available.

One methodological comment I might make was that it was somewhat unclear to me why the authors would sonicate their DNA prior to subjecting it multiplex PCR, unless this was simply done to simulate a fragmented FFPE sample? Regardless, since the authors used sonicated fragmented DNA as their input, and also subjected the final PCR product (amplified at a relatively low 17 cycles) to ligation based library preparation methods, they may also want to assess separately the extent to which the off target reads may actually be sheared genomic DNA coming thru in the library prep process. Running a sheared DNA sample without any primers through the same PCR, cleanup and library prep process may help to assess the extent to which the off target reads are coming from passenger DNA.

Some specific points. Fig S6 appears to be mislabelled, as the text says that on target amplicons were 90-120bp in size, but in the figure the pink read counts at this size are labelled as being dimer. Figure S11 may also have the same problem, although for this figure I'm not certain as the text description of results I found a bit unclear.

As well, their method looks to go through a substantial number of iterations (40000+ from the sounds of it) to generate their final panel. It would also be useful for them to include a table or figure which gives a sense of the total computation time it takes to perform this process, with single and multi core times for the different sized panels they designed (assuming their code has been parallelized).

I also found the methods description of the PCR conditions to be a bit lacking. A more detailed description which includes the salt/Mg concentration used, dNTP concentrations, cycling conditions and times, and final primer concentration (expressed as the final concentration of each primer in the final pool) is required to be able to easily replicate the methodology.

Reviewer #3:

Remarks to the Author:

This paper is a study on how to design multiplex PCR for a large number of genes. Simultaneous amplification of a large number of genes is an area of increasing academic and social need due to the increasing use of amplicon sequencing and the advancement of qPCR technology.

It is well known that the obstacle to multiplex PCR is dimer formation, and the development of an algorithm that suppresses dimer formation and increases the number of primer pairs for multiplex PCR is expected to make the design more difficult.

A basic algorithm for suppressing dimer formation in primer design has already been proposed in the following paper.

Shen, Zhiyong, et al. "MPprimer: a program for reliable multiplex PCR primer design." *BMC bioinformatics* 11.1 (2010): 1-7.

Johnston, Andrew D., et al. "PrimerROC: accurate condition-independent dimer prediction using ROC analysis." *Scientific reports* 9.1 (2019): 1-14.

However, as described in the following paper, the computational method of reducing the number of dimer combinations extends the computation time in a series as the number of target genes increases, and no algorithm has been proposed for such a large number of target genes as described in this study.

Rachlin, John, et al. "Computational tradeoffs in multiplex PCR assay design for SNP genotyping." *BMC genomics* 6.1 (2005): 1-11.

This study solves the problem of computational time by using a simpler function called the Badness function as an indicator. In addition, by releasing the source code of the program, it enables many researchers to design multiplex PCR primers for a large number of genes.

From these points, even though there is no novelty in the basic concept, it is considered to meet the criteria for the Journal (*Nature Communication*) in two aspects: broad practicality and convenience for many researchers.

However, it is considered that the following points need to be explained and improved on the revision.

Introduction:

First paragraph:

The authors cite the Amplicon sequence of NGS and qPCR as the background for this study, but the reference for qPCR shows the simultaneous amplification of only 5-6 different viruses, which is inappropriate as a reference for this study, which aims for the simultaneous amplification of more than 60 viruses.

Third paragraph:

The authors should refer to previous papers where the series increase in computation time associated with N-plex PCR primer design.

Results:

1. Primer candidate generation:

Please show the specific bp and number of designs for the first primer setting position. The authors should provide specific data on the process of predicting $\Delta G = -11.5$ kcal/mol in the supplement.

3. Evaluation of Loss function:

For equation (1), please show that Fig. 1 is referenced.

In Fig. 1, primers are selected randomly for a and b, so combinations of forward and forward are also selected. Describe how to select forward-reverse in the end of the primer selection.

For equation (2), it is written "defined as follows", but the meaning is not clear without looking at Fig2 b. Please note that Fig2b shows the meaning of each symbol.

The reference in the thermodynamic parameter would be outdated. There are many of the latest papers in which the latest calculation of Gibb's energies, and the authors should refer to them.

It is written that Enthalpy and Entropy do not hold at high temperatures, but it should be stated whether 60°C used in this study is such a high temperature.

In the last paragraph of this section, the author describes the calculation time of the loss function as $O(N^2 \cdot P^2)$. While the next section states that it could be reduced to $O(N \cdot P^2)$. The relationship between the two description should be carefully explained.

Generate a temporary primer set T based on Sg:

The authors have performed multiple random generation of Sg and calculation of the loss function at the same time. Usually, in such calculations, the optimal solution is obtained by reducing the error value. Nevertheless, the method for obtaining the optimal solution while performing simultaneous calculations is not understood even after reading the next section. Please explain in detail the simultaneous calculation and reduction of errors.

6. Repeat steps 4 and 5:

Please explain the criteria for stopping repetition in detail.

Design and Experimental Evaluation:

Please clarify whether the authors did or did not improve SADDLE by referring to the NGS results in PS1-3.

Accuracy of the Dimerization Prediction:

The list of predicted and actual dimers and their frequencies should be presented in full in the Supplement for researchers to consider using and improving this method.

Fig4de The large difference between the predicted and actual dimer may contradict the core of the significance of this study. Please explain clearly that primer set selection based on Budness function is meaningful even if the prediction is different from the actual result.

Discussion:

First paragraph:

In this study, researchers will be able to design multiplex PCR for a large number of genes with SADDLE. However, in such studies, multiplex PCRs are often designed by selecting genes for which primer sets can be created. In contrast, many researchers have a set of genes they want to run PCR, and want to design a multiplex PCR for their set. The author would like to explain to what extent SADDLE can be applied to the set of genes that the researcher wants.

Second paragraph:

Please refer to the appropriate references.

Third paragraph:

Since the authors have real data from NGS, the authors should suggest how to improve the SADDLE they initially designed with reference to the NGS results. In doing so, it is recommended to refer to the research in machine learning of nucleotide interaction including dimers (Kayama et al. Scientific Reports, 2021).

Fourth paragraph:

The authors propose qPCR as a lower cost method of analysis than NGS, but no references cited. The authors should provide the references for such a qPCR method without NGS. If this cannot be shown, this description should be deleted.

Code Availability:

Please be available for codes to review at <https://github.com/NinaGXie>.

Response to reviewers

Reviewer #1

The authors describe a computational implementation for multiplex PCR design they term SADDLE, and use this methodology to generate a series of multiplex panels for a variety of PCR applications (e.g., NGS sequencing, sanger, qPCR detection).

The methodology is described in computational and mathematical terms. However, the key application seems to centre on what the authors term their Badness function, which thematically appears to be based predominantly on 5mr base pair pattern matching for all primer pairs within a pool; followed by bootstrapping randomly sampled primers to maximize dangling 3' ends of the predicted interactions, which the authors believe minimizes dimer formation.

Overall the manuscript is very nicely laid out, and the authors have attempted to characterize their platform across a variety of relevant, clinical situations (e.g., FFPE and fusion events). The qPCR multiplex assay for fusion detection in particular I thought was a very nice clinical application for rapid detection of fusion events from the cDNA of cancer patients.

We thank the reviewer for the kind comments.

Methodologically, the authors have characterized their system quite well, and I commend them for the amount of detail as well as the attempt to make their method clinically relevant; however, I'm not entirely convinced of the novelty of the proposed work. Although the mathematical description and validation work is well described, my assessment is that the methodology can mostly be reduced to the pattern matching loop that iterates though all possible primer pairs, followed by weighting for nonbinding events at the 3' end of the primer, and neither of these solutions would be particularly new to the field.

The idea of pattern matching for primer design is indeed applied in several studies:

Shen, Z., Qu, W., Wang, W., Lu, Y., Wu, Y., Li, Z., ... & Zhang, C. MPprimer: a program for reliable multiplex PCR primer design. BMC bioinformatics, 11(1), 1-7 (2010).

Lu, J., Johnston, A., Berichon, P., Ru, K. L., Korbie, D., & Trau, M. PrimerSuite: a high-throughput web-based primer design program for multiplex bisulfite PCR. *Scientific reports*, 7(1), 1-12 (2017).

Wingo, T. S., Kotlar, A., & Cutler, D. J. MPD: multiplex primer design for next-generation targeted sequencing. *BMC bioinformatics*, 18(1), 1-5 (2017).

Kechin, A., Borobova, V., Boyarskikh, U., Khrapov, E., Subbotin, S., & Filipenko, M. NGS-PrimerPlex: High-throughput primer design for multiplex polymerase chain reactions. *PLoS Computational Biology*, 16(12), e1008468 (2020).

Yet none of these studies managed to scale up to more than 50-plex PCR. This is mainly because of the computational complexity of predicting the overall dimerization of a multiplexed PCR. Our study took advantage of the badness function using the hash table, enabled the optimization of primer dimer in a time-efficient way.

Furthermore, the rather significant off-target amplification of their panels substantially limits the impact of their methodology.

The off-target effect is assumed to be caused by primers binding to undesired locations of the genome, while primer dimerization is caused by primer-primer interactions. In this study, we mainly focused on predicting primer dimerization, and we will try to optimize off target effect in future studies.

While the authors are transparent in the off-target effects observed in their data, I would have to be somewhat critical of the comment made that their pipeline is not optimized to control for off target effects, particularly since their methodology would be most readily compared to the Ampliseq application, which I would generally characterize as the market leader in this capacity. To draw a comparison, Ampliseq's CCP cancer panel I can state from experience has multiplex pools with 8000 primers in single tube and achieves an on-target rate in the high 90% with virtually no primer dimer formation whatsoever. The Ampliseq transcriptome and exome panels are even higher (40,000 primer pairs and 200,000 respectively, if memory serves), and the

presence and success of this commercial solution therefore takes away from the impact and relevance of the authors work rather substantially, unfortunately.

We thank the reviewer for mentioning Ampliseq and it is important for us to clarify on this. We performed a side-by-side comparison between SADDLE and Ampliseq by ordering a customized panel from Illumina covering the same 96 genomic loci as SADDLE primer set 1~3. The detailed results can be found in Supplementary Section 10.

Among the 96 loci, Ampliseq design failed on 1 locus, and 5 of them are covered by 2 primer pairs each, resulting a total of 100 primer pairs. In addition, Ampliseq primers were delivered in 2 primer pools (50 primer pairs in each pool). Library preparation was performed according to the public standard Ampliseq protocol, and NGS data (MiSeq v2 pair-end) was analyzed using the same bioinformatics pipeline as SADDLE.

We agree with the reviewer that barley any dimer can be observed for the libraries prepared with Ampliseq's standard protocol, however in our case, 30% of NGS reads could not be aligned to targeted loci, giving 70% final on-target rate. Based on the Bioanalyzer results (Fig. S27), we believe that those unmapped reads are from non-specific amplicons longer than the on-target amplicons.

A more direct comparison with SADDLE PS3 (96 primer pairs in one pool) is mixing the Ampliseq's primer panel pool 1 and panel pool 2 before the first multiplex PCR (100 primer pairs in one primer pool), and with no size selection (same as SADDLE experiment), the short reads (< 60 bp) ratio of this library is 6.4%.

Based on the patent (Leamon, J., Andersen, M., & Thornton, M. U.S. Patent No. 9,957,558. Washington, DC: U.S. Patent and Trademark Office (2018)) and experiment protocol, Ampliseq has a unique NGS library preparation workflow where they 1) introduce an additional step that use FuPa enzyme mixture to digest primer dimers by cleaving the modified bases in primers, and 2) include two separate steps of SPRI size selection using AMPure beads. Although Ampliseq could achieve higher multiplexity while maintaining a high on-target rate, we reckon it as more of a unique NGS library preparation method than a conventional multiplexed PCR.

On the other hand, SADDLE is a computational workflow to design a set of primers with low dimer formation, which would work on any multiplexed PCR with no additional enzymatic clean up needed or size selection. It is sometimes difficult/impossible for users to apply enzyme digestion in any multiplex PCR, since this could limit the experimental condition to certain chemistry. With SADDLE software, researcher can directly use the designed primer sequences to any application, including qPCR and customized NGS library preparation. Furthermore, FuPa enzyme mixture used in Ampliseq's protocol digests modified bases in the primers, leading to truncations at the 5' end and/or 3' end of the amplicons (Fig. S29). These truncations might prevent FuPa's application on multiplexed PCR with unique molecular identifier (UMI), since UMIs are usually attached to the 5' end of the primers.

Finally, Ampliseq's protocol includes two separate steps of size-selection (1st time 1.0x, and 2nd time 0.5x), reducing dimers and long non-specific amplicons, while no size-selection is included in our NGS workflow as we are trying to observe as much as dimer species as possible. However, we still observed ~70% on-target rate and 4% dimer rate in primer set 3 (Fig. 3A). And we believe that if additional size-selection is applied, there would be less dimer compared to current results.

As such, while the authors have written a very nice paper and done a lovely job characterizing and applying their work, I'm not entirely convinced as to the novelty and overall relevance of their methodology to the field, particularly given these limitations and the other commercial solutions currently available.

As discussed above, SADDLE is a universal primer design algorithm to all multiplexed PCR, while Ampliseq is a unique library preparation method that cannot be readily applied elsewhere. Although we believe combining SADDLE and Ampliseq's chemistry could yield the best results, we think that Ampliseq is not a direct comparison or a substitute to our method.

One methodological comment I might make was that it was somewhat unclear to me why the authors would sonicate their DNA prior to subjecting it multiplex PCR, unless this was simply done to simulate a fragmented FFPE sample? Regardless, since the authors used sonicated fragmented

DNA as their input, and also subjected the final PCR product (amplified at a relatively low 17 cycles) to ligation-based library preparation methods, they may also want to assess separately the extent to which the off target reads may actually be sheared genomic DNA coming thru in the library prep process. Running a sheared DNA sample without any primers through the same PCR, cleanup and library prep process may help to assess the extent to which the off target reads are coming from passenger DNA.

We thank the reviewer for pointing out a possible source of off target reads. We performed the experiment as the reviewer suggested.

We used sonicated DNA as to simulate fragmented FFPE sample and cell-free DNA (peak length usually ~150 bp). Sonicated DNA alone without any primers (No Primer Control (NPC)) was run through the same library preparation workflow, and we measure the Ct values of both NPC library (Ct = 20) and normal library (Ct = 9) before the final Index PCR. The Ct difference suggests $2^{11} = 2048$ difference in molecules. Our hypothesis is that after 17 cycles of denature-annealing in the first multiplexed PCR step, almost all non-targeted sonicated DNA fragments are single stranded, and it is almost impossible for them to perfectly hybridize again considering their ultra-low concentration. Therefore, almost no non-targeted sonicated DNA fragments could be ligated with NGS adapters or observed in the final library.

Some specific points. Fig S6 appears to be mislabelled, as the text says that on target amplicons were 90-120bp in size, but in the figure the pink read counts at this size are labelled as being dimer. Figure S11 may also have the same problem, although for this figure I'm not certain as the text description of results I found a bit unclear.

We thank reviewer for the comments. The labels are changed to correct form. We'd like to clarify that no primer concentration modification was done in Fig. S11 (30nM/primer, a total of 5.76 uM), and we only did one round of primer concentration tuning to optimize the performance, as shown in Fig. 3a (average primer conc is 45 nM/primer).

As well, their method looks to go through a substantial number of iterations (40000+ from the sounds of it) to generate their final panel. It would also be useful for them to include a table or figure which gives a sense of the total computation time it takes to perform this process, with single and multi-core times for the different sized panels they designed (assuming their code has been parallelized).

We appreciate the reviewer's comments and calculated the computation time. Our computation time for a 96-plex panel design is about 490 seconds (single core) for 60,000 iterations on a conventional laptop; and for a 384-plex panel design is about 2750 seconds (single core) for 60,000 iterations on a conventional laptop. We added the time to Fig.2(d).

I also found the methods description of the PCR conditions to be a bit lacking. A more detailed description which includes the salt/Mg concentration used, dNTP concentrations, cycling conditions and times, and final primer concentration (expressed as the final concentration of each primer in the final pool) is required to be able to easily replicate the methodology.

Please find detailed library preparation protocols including salt, dNTP concentration and cycling conditions in Supplementary Section 3. We also include the Ampliseq experiment protocol in the Supplementary Section 10.

Primer sequence and concentration in PS1-PS3, and Ampliseq amplicon sequence are all included in the Supplementary excel file.

Reviewer #2

This paper is a study on how to design multiplex PCR for a large number of genes. Simultaneous amplification of a large number of genes is an area of increasing academic and social need due to the increasing use of amplicon sequencing and the advancement of qPCR technology.

It is well known that the obstacle to multiplex PCR is dimer formation, and the development of an algorithm that suppresses dimer formation and increases the number of primer pairs for multiplex PCR is expected to make the design more difficult.

A basic algorithm for suppressing dimer formation in primer design has already been proposed in the following paper.

Shen, Zhiyong, et al. "MPprimer: a program for reliable multiplex PCR primer design." *BMC bioinformatics* 11.1 (2010): 1-7.

Johnston, Andrew D., et al. "PrimerROC: accurate condition-independent dimer prediction using ROC analysis." *Scientific reports* 9.1 (2019): 1-14.

However, as described in the following paper, the computational method of reducing the number of dimer combinations extends the computation time in a series as the number of target genes increases, and no algorithm has been proposed for such a large number of target genes as described in this study.

Rachlin, John, et al. "Computational tradeoffs in multiplex PCR assay design for SNP genotyping." *BMC genomics* 6.1 (2005): 1-11.

This study solves the problem of computational time by using a simpler function called the Badness function as an indicator. In addition, by releasing the source code of the program, it enables many researchers to design multiplex PCR primers for a large number of genes.

We appreciate the reviewer for the review of previous studies. These papers have been cited in the introduction section (line 16-17). The corresponding main text is copied below (Introduction paragraph 2)

“To the best of our knowledge, existing multiplex primer design algorithm never exceeded 50 primer pairs in one tube [11–14]. This is mainly due to the high computational cost when the number of primers increases [15].”

From these points, even though there is no novelty in the basic concept, it is considered to meet the criteria for the Journal (Nature Communication) in two aspects: broad practicality and convenience for many researchers.

We thank the reviewer for the kind comments.

However, it is considered that the following points need to be explained and improved on the revision.

Introduction:

First paragraph:

The authors cite the Amplicon sequence of NGS and qPCR as the background for this study, but the reference for qPCR shows the simultaneous amplification of only 5-6 different viruses, which is inappropriate as a reference for this study, which aims for the simultaneous amplification of more than 60 viruses.

We thank the reviewer for the comments. We tried to search for a study with much higher plex number qPCR test. However, the detection of highly multiplex variants is not frequent in a qPCR reaction because the number of fluorescence channels available in a qPCR instrument is ≤ 6 . (With one channel used for each single-plex variant, usually simultaneous variant detection is limited to 6-plex). And in our own design, we didn't use the qPCR to identify the exact variant, but use Sanger sequencing for further variant confirmation, therefore we can achieve a much higher plex number.

Third paragraph:

The authors should refer to previous papers where the series increase in computation time associated with N-plex PCR primer design.

Citation was added in the second paragraph of introduction (line 17).

“In contrast, relatively little systematic work has been reported on computational approaches to minimizing the formation of primer dimers in the first place. This is mainly due to the high computational cost when the number of primers increases [15].”

Results:

1. Primer candidate generation:

Please show the specific bp and number of designs for the first primer setting position.

The authors should provide specific data on the process of predicting $\Delta G = -11.5$ kcal/mol in the supplement.

The specific bp and number of designs varies depending on 1) accepted range of amplicon length, 2) the length of the pivot sequence and 3) the context of target of interest (GC ratio, etc.). We will generate as many primer pair candidates as possible if they satisfy the constraints. As shown in Fig. 2(a), for example, if the designed Maximum Amplicon Length (MAL) is 121 bp, and our pivot sequence is a single base SNP/mutation, the first start position of the forward primer candidates will be 60 bp upper stream of the pivot sequence.

And please find detailed data of how we calculated the ΔG of primer binding in Supplementary Section 9.

3. Evaluation of Loss function:

For equation (1), please show that Fig. 1 is referenced.

We referenced the Fig.1 in equation (1).

In Fig. 1, primers are selected randomly for a and b, so combinations of forward and forward are also selected. Describe how to select forward-reverse in the end of the primer selection.

Forward primer candidates and reverse primer candidates are first paired combinatorially as primer pair candidates. (For example, if we generate M fP and N rP, together we have $M*N$ primer pairs as primer pair candidate.) For each DNA target, a random primer pair candidate is added into the initial primer set. During each iteration, a primer pair candidate is replaced with another primer pair candidate of the same DNA target, and the Loss is calculated for the original primer set and the new primer set. This process is also described in Fig. 1 and page 3 under "Primer candidate generation".

"In the implementation of SADDLE, primer candidates can be treated as individual primers or as primer pairs. Our specific implementation treats primers as pairs, so we next combinatorially generate all candidate primer pairs for an DNA target, in order to better constrain the distribution of amplicon lengths.

Any candidate primer pairs that generate amplicons with length exceeding our maximum amplicon length or below our minimum are removed."

For equation (2), it is written "defined as follows", but the meaning is not clear without looking at Fig2 b. Please note that Fig2b shows the meaning of each symbol.

We rewrote the equation 1 and 2 with clearer symbol name and re-plot Fig.2b. Please find more detailed description in the updated "3. Evaluation of Loss function $L(S)$ on S_0 " at page 3.

The reference in the thermodynamic parameter would be outdated. There are many of the latest papers in which the latest calculation of Gibb's energies, and the authors should refer to them.

We thank the reviewer for the comments and added two more citations in the revised manuscript (line 110). We keep Santa Lucia (2004) paper as reference as from which the parameters we referred to in our research. Please find below the articles we referenced:

"[17] Zacharias, M. Base-pairing and base-stacking contributions to double-stranded DNA formation. J Phys Chem B, 124(46), 10345-10352 (2020).

[18] Huguet, J. M., Ribezzi-Crivellari, M., Bizarro, C. V., & Ritort, F. Derivation of nearest-neighbor DNA parameters in magnesium from single molecule experiments. *Nucleic Acids Res*, 45(22), 12921-12931(2017)."

It is written that Enthalpy and Entropy do not hold at high temperatures, but it should be stated whether 60°C used in this study is such a high temperature.

In the previous study, we measured the dG° at different temperatures ranging from 20° to 70°, and 60° is a relatively high temperature.

In the last paragraph of this section, the author describes the calculation time of the loss function as $O(N^2 \cdot P^2)$. While the next section states that it could be reduced to $O(N \cdot P^2)$. The relationship between the two descriptions should be carefully explained.

We added more detailed and precise description in the updated Fig.2b and "3. Evaluation of Loss function $L(S)$ on S_0 " at page 3. We also added equation 3 to describe how hash table accelerates Loss calculation. Please find the paragraph where time complexity is discussed below.

"The evaluation of the Badness function is single largest component of software runtime cost, due to the large number of times the Badness function will be evaluated. For a primer that is 25 nt in length, there are 22 subsequences of length 4, 21 subsequences of length 5, etc. Evaluation of Badness for a single primer pair would thus have time complexity of $O(P^3)$, where P is the length of each primer (Eqn. 2). In our specific implementation, subsequence length len also has a maximum of 8 nt, decreasing the time complexity to $O(P^2)$. Naively, evaluation of the Loss $L(S)$ of the whole primer set would have time complexity of $O(N^2 \cdot P^2)$ (Eqn. 1). However, due to the additive nature of subsequence components to the overall Badness function, we implement more rapid Badness evaluation by using a hash table [20], as shown in Eqn. 3, where H is the hash table, s is a subsequence of the primer set, d is the distance to 3' ends of each occurrence of s , and $revcomp$ is a function that converts s to its reverse complement sequence (Fig. 2b). The time complexity to set up the hash table is $O(N \cdot P)$ to calculate the hash value for each unique subsequence in the primer set, and the time complexity to evaluate the $L(S)$ by stepping through all subsequences of all primers is also $O(N \cdot P)$ (Eqn. 3). Consequently, the overall time complexity of evaluating $L(S)$ is $O(N \cdot P)$ for all primers in S ."

Generate a temporary primer set T based on Sg:

The authors have performed multiple random generation of Sg and calculation of the loss function at the same time. Usually, in such calculations, the optimal solution is obtained by reducing the error value. Nevertheless, the method for obtaining the optimal solution while performing simultaneous calculations is not understood even after reading the next section. Please explain in detail the simultaneous calculation and reduction of errors.

During each generation, a primer pair in Sg is replaced with another primer pair of the same DNA target, and the “mutated” primer set is denoted as T. Since the hash table of Sg is already set up, the hash table of T does not need to be set up again. Instead, it only needs to be modified based on the hash table of Sg according to the replaced primers. Therefore, the Loss of the mutated primer set T $L(T)$ can be rapidly evaluated. $L(T)$ is then compared with $L(Sg)$ to determine whether the primer pair replacement is kept or not. Please also find corresponding text at the beginning of “5. Evaluate $L(T)$ and set $Sg+1$ to be either T or Sg”.

“The Loss of temporary primer set T can be evaluated significantly faster than the initial evaluation of $L(S_0)$, because the hash table only need to be modified based on the changed primers.

We next compare the value of $L(T)$ vs. $L(Sg)$. If $L(T)$ is smaller than $L(Sg)$, then the primer pair change was an improvement and accepted, so $Sg+1$ is set to T. If $L(T)$ is larger than $L(Sg)$, the change was detrimental, but we will still accept the change with a certain probability, as part of the simulated annealing algorithm [21].”

6.Repeat steps 4 and 5:

Please explain the criteria for stopping repetition in detail.

During the simulated annealing process, the initial “temperature” for simulated annealing is 1000, and it decreases 100 per step, with 2000 iterations in each step. Once the temperature reaches zero, it no longer decreases and there are another 40000 iterations before the optimization stops. Those parameters can be changed, and user can visualize the optimization process by plotting the Loss vs. number of iterations to make sure the Loss reaches plateau.

Design and Experimental Evaluation:

Please clarify whether the authors did or did not improve SADDLE by referring to the NGS results in PS1-3.

We haven't improved SADDLE by NGS results of PS1-3, however we planned to optimize it using machine learning in our future work. And we also mentioned this point in the discussion.

Accuracy of the Dimerization Prediction:

The list of predicted and actual dimers and their frequencies should be presented in full in the Supplement for researchers to consider using and improving this method.

We added the predicted dimers and actual dimers in supplementary excel file. Please find data in attached file: **SADDLE_Supplementary_Badness_score** for predicted dimer, and **SADDLE_Supplementary_NGS_data** for actual dimer.

Fig4de The large difference between the predicted and actual dimer may contradict the core of the significance of this study. Please explain clearly that primer set selection based on Badness function is meaningful even if the prediction is different from the actual result.

The AUROC score of predicting primer dimerization with the Badness function is 0.9577 (Fig. 4a-c), which is considered acceptable in most machine learning and optimization problems. On the other hand, predicting the dimerization of each individual primer pair accurately is not the most important purpose of this study. We mainly aim to predict the overall level of dimerization of the whole primer set, referred as the Loss function. From Fig. 2d-e, the Loss of PS1-3 reflects the total amount of dimer formed in the primer set.

Furthermore, the main difference between the most predicted dimer and experimental dimer is that the most experimental dimers all have perfect matching bases at 3'end; this could be caused by the DNA polymerase Vent(exo-) used in the PCR: We used vent(exo-) to improve the amplification of several high-GC ratio targets. However, vent(exo-) does not have the 3'→5' exonuclease activity; therefore, dimers that have perfect binding in the middle will not be cleaved

and extended. If the researcher use polymerase with 3'→5' exonuclease (e.g., Phusion), the actual dimer formation will be different, the optimization of the penalty at different positions can be modified accordingly to different polymerase in the future.

Discussion:

First paragraph:

In this study, researchers will be able to design multiplex PCR for a large number of genes with SADDLE. However, in such studies, multiplex PCRs are often designed by selecting genes for which primer sets can be created. In contrast, many researchers have a set of genes they want to run PCR and want to design a multiplex PCR for their set. The author would like to explain to what extent SADDLE can be applied to the set of genes that the researcher wants.

We thank reviewer for the comments. SADDLE currently accepts the inputs as a fasta file that includes any DNA sequences of interest and a specific position (pivot sequence) the researcher wants to include in the amplicon (SNPs, INDEL, etc).

The DNA sequences can be fetched from the NCBI database by their gene names and coordinates, which we will also include in future versions of SADDLE as well.

Other detailed instructions about how to use SADDLE are provided in program package.

Second paragraph:

Please refer to the appropriate references.

Please find more references in the second paragraph (line 311-320).

“[33] Bewicke-Copley, F., Kumar, E. A., Palladino, G., Korfi, K., & Wang, J. Applications and analysis of targeted genomic sequencing in cancer studies. *Comput Struct Biotechnol J*, 17, 1348-1359 (2019).

[34] Siravegna, G., Mussolin, B., Venesio, T., Marsoni, S., Seoane, J., Dive, C., ... & Bardelli, A. How liquid biopsies can change clinical practice in oncology. *Ann Oncol*, 30(10), 1580-1590. (2019).

[35] Hardy, T. The role of prenatal diagnosis following preimplantation genetic testing for single-gene conditions: A historical overview of evolving technologies and clinical practice. *Prenat Diagn*, 40(6), 647-651 (2020).”

Third paragraph:

Since the authors have real data from NGS, the authors should suggest how to improve the SADDLE they initially designed with reference to the NGS results. In doing so, it is recommended to refer to the research in machine learning of nucleotide interaction including dimers (Kayama et al. *Scientific Reports*, 2021).

We appreciate the reviewer’s suggestion and added the reference in our discussion paragraph 3 (line 330).

“Current Loss function can be further improved based on NGS data and other methods including machine learning [36].

[36] Kayama, K., Kanno, M., Chisaki, N., Tanaka, M., Yao, R., Hanazono, K., ... & Endoh, D. Prediction of PCR amplification from primer and template sequences using recurrent neural network. *Sci Rep*, 11(1), 1-24 (2021).”

Fourth paragraph:

The authors propose qPCR as a lower cost method of analysis than NGS, but no references cited. The authors should provide the references for such a qPCR method without NGS. If this cannot be shown, this description should be deleted.

We thank reviewer for the comments and added one citation in our manuscript (line 332).

“In medical and research applications where the cost of NGS cannot be economically justified [37].

[37] Cervena, K., Vodicka, P., & Vymetalkova, V. Diagnostic and prognostic impact of cell-free DNA in human cancers: Systematic review. *Mutat Res*, 781, 100-129 (2019) ”

Code Availability:

Please be available for codes to review at <https://github.com/NinaGXie>.

Please find updated information in our Code Availability and data availability.

'Data Availability. Raw NGS data is available at <https://doi.org/10.6084/m9.figshare.16944154>.

Code Availability. The MATLAB code and Python code for NGS data processing are available at <https://github.com/NinaGXie/SADDLE>. The MATLAB code used for multiplex PCR primer algorithm is available upon request under NDA for academic lab.'

We'd like to clarify that as we have a pending patent on the SADDLE described in this work, the design code will be available upon request under NDA for academic labs.

Reviewers' Comments:

Reviewer #2:

Remarks to the Author:

In responding to my original comments, the authors are broadly attempting to position their application as being able to deliver single pool reactions. While I appreciate this perspective I believe the overall concerns I originally stated still remain.

First, the authors are incorrect in their assessment of the Ampliseq application. While their custom Ampliseq panel may have arrived as two 50 plex pools, this is not because Ampliseq is unable to design a single pool. It arrived as two pools because when Ampliseq designs a panel, proximal assays (for example a large exon requiring two or more assays to cover) are separated into 2 pools to prevent them from interacting in a way that generates larger product formation. As such, the authors claim that mixing the two Ampliseq pools together make the assay analogous to their SADDLE assay misses the point of why the Ampliseq pools were separate in the first place.

Building on this point, the authors claim that the Ampliseq panel resulted in 30% off target reads, but their SADDLE panel also appears to have the same off target rate of 30% (figure 3a).

Returning to my original reviewers comment, Ampliseq can also build highly multiplexed panels of 200000 primer pairs, which again rebuts the authors claims that no other solution has achieved a plex greater than 50 assays.

As such, while I appreciate the authors comments, the thrust of their rebuttal is predominantly an argument against the primary review comments, as opposed to substantive methodological improvements to their methodology or compelling rhetorical statements.

Overall, the main argument the authors take regarding impact and novelty is that their method can utilize non-modified oligonucleotides, as opposed to (for example) Ampliseq, which the authors are correct in asserting that modified based and enzymatic treatment are both part of this workflow.

On this regard (using unmodified oligos) as well as their claims of advancing multiplexing are rather weak, as one of the papers they cite in their rebuttal (Johnston, "PrimerROC", SciReports) achieved a 63 plex compared to the authors 96, which I would classify as an incremental advancement.

Arguments against size selection and bead cleanup I also find somewhat specious, given that this step is ubiquitous in all NGS labs and extremely cheap and quick. Indeed, for cluster generation on Illumina platforms cleanup is absolutely required to remove adaptor dimer and unincorporated primers, as both these species compete during cluster generation. As such, claims their method does not require this is somewhat dubious.

As such, the fundamental claim remains whether their 3' end match function significantly advances the field. Again, the authors have done a very detailed job and should be commended, but their solution lags behind the market leader in terms of dense multiplexing (leaving aside the question of modified primers and enzymatic treatments); as well as novelty, as per the papers the authors themselves cite in their response.

While the manuscript is nicely laid out, the decision to publish would mostly be reduced down to the editors decision, as I personally believe their method is not a substantive advancement for the field.

Reviewer #3:

Remarks to the Author:

The authors responds appropriately to all my comments. The revised manuscript is also written properly. I conclude that this paper should be accepted.

However, please consider the following as a minor revision.

1 In the ROC curves of Fig.4B and Fig.S21bd, the authors set the horizontal axis to Specificity, but usually 1-Specificity is set for X-axis. Please consider modifying x-axis to 1-Specificity for the reader's understanding.

2 I cannot find a description about the actual calculation platform (CPU, OS, etc.) in the main body of the paper and Supplement. On the other hand, in SADDLE.m in Supplementary Software, the time required for calculation is "estimated run time: 494 seconds for 96 DNA targets (Intel Xeon W-11955M Processor), 2754 seconds for 384 DNA targets (Intel Xeon W-11955M Processor).)". The calculation environment and required time are necessary information for researchers who want to use SUDDLE for research, so please consider to describe the calculation environment and required time on authors' environment in the main body or Supplement.

3 The source code currently released on Github is simplified compared to Supplementary Software. When publishing a paper, the softwares in "Supplementar software" for the review should be opened on Github.

Reviewer #2 (Remarks to the Author):

In responding to my original comments, the authors are broadly attempting to position their application as being able to deliver single pool reactions. While I appreciate this perspective I believe the overall concerns I originally stated still remain.

We thank the reviewer for the comments. We would like to clarify that, in our original response, our major claim is SADDLE is universal to all PCR (e.g., qPCR, PCR with unique molecular identifiers) since it does not need modified bases or enzymatic treatment, as opposed to Ampliseq. Please find a part of our response below.

“On the other hand, SADDLE is a computational workflow to design a set of primers with low dimer formation, which would work on any multiplexed PCR with no additional enzymatic clean up needed or size selection. It is sometimes difficult/impossible for users to apply enzyme digestion in any multiplex PCR, since this could limit the experimental condition to certain chemistry. With SADDLE software, researcher can directly use the designed primer sequences for any application, including qPCR and customized NGS library preparation.

Furthermore, FuPa enzyme mixture used in Ampliseq’s protocol digests modified bases in the primers, leading to truncations at the 5’ end and/or 3’ end of the amplicons (Fig. S29). These truncations might prevent FuPa’s application on multiplexed PCR with unique molecular identifier (UMI), since UMIs are usually attached to the 5’ end of the primers.”

First, the authors are incorrect in their assessment of the Ampliseq application. While their custom Ampliseq panel may have arrived as two 50 plex pools, this is not because Ampliseq is unable to design a single pool. It arrived as two pools because when Ampliseq designs a panel, proximal assays (for example a large exon requiring two or more assays to cover) are separated into 2 pools to prevent them from interacting in a way that generates larger product formation. As such, the authors claim that mixing the two Ampliseq pools together makes the assay analogous to their SADDLE assay misses the point of why the Ampliseq pools were separate in the first place.

We thank the reviewer for the comment. We would like to clarify that our customized Ampliseq panel targets the same 96 loci as our 96-plex panel, and each one of them is from different cancer related gene (96 genes). Therefore, there is no overlapping between the 96 loci. We originally placed the Ampliseq order in “hotspot” configuration (single pool). However, customer service from Illumina informed us that the design failed and asked us to try the “gene” configuration (multiple pools) instead. As a result, we received two pools of primers instead of one.

We performed the experiments and followed Ampliseq’s library preparation protocol for two primer pools and presented the results in our Supplementary Information. Still, we hypothesized that merging the two pools is still legitimate since the targeted 96 loci are separated. Moreover, merging the two pools could make a fairer comparison between Ampliseq and SADDLE, since our 96-plex panel only has one pool.

Please find related text in our manuscript or our original response below.

“We performed a side-by-side comparison between SADDLE and Ampliseq by ordering a customized panel from Illumina covering the same 96 genomic loci as SADDLE primer set 1~3.”

“We first used SADDLE to optimize the design of a 96-plex primer set, each amplicon target one arbitrarily selected exon of a different cancer-related gene”

Building on this point, the authors claim that the Ampliseq panel resulted in 30% off target reads,

but their SADDLE panel also appears to have the same off target rate of 30% (figure 3a).

Returning to my original reviewers comment, Ampliseq can also build highly multiplexed panels of 200000 primer pairs, which again rebuts the authors claims that no other solution has achieved a plex greater than 50 assays.

We thank the reviewer for pointing out our inaccurate argument. We'd like to restate it as: no other solution has achieved a plex greater than 70 without using modified bases or enzymatic treatment.

As such, while I appreciate the authors comments, the thrust of their rebuttal is predominantly an argument against the primary review comments, as opposed to substantive methodological improvements to their methodology or compelling rhetorical statements.

Overall, the main argument the authors take regarding impact and novelty is that their method can utilize non-modified oligonucleotides, as opposed to (for example) Ampliseq, which the authors are correct in asserting that modified based and enzymatic treatment are both part of this workflow.

On this regard (using unmodified oligos) as well as their claims of advancing multiplexing are rather weak, as one of the papers they cite in their rebuttal (Johnston, "PrimerROC", SciReports) achieved a 63 plex compared to the authors 96, which I would classify as an incremental advancement.

We would like to clarify that we can build a 384-plex panel while maintaining a low fraction of dimer reads (1%). We extensively studied three 96-plex panels as proof of concept, we also presented our results of a 384-plex panel both in the main text (Fig. 3f) and supplementary section S6, with a low dimer fraction and high on-target rate. We think it's still an advancement compared to the existing 63-plex. Though we'd be happy to perform much higher plex experiments, due to our lab's limited budget, we weren't able to do it.

Please find a detailed description in our abstract and Fig. 3f.

"To demonstrate the scalability of SADDLE, we next designed and tested a 384-amplicon panel comprising 768 primers. Due to the high cost of primer synthesis for this large panel, we only experimentally tested the final primer set design. Surprisingly, the observed Dimer fraction was only 1% for this library, using an input of 40 ng sheared NA18562 genomic DNA (Fig. 3f). Roughly 56% of the reads were Non-specific amplicons, resulting in an NGS library on-target rate of 43% (Supplementary section S6)."

Arguments against size selection and bead cleanup I also find somewhat specious, given that this step is ubiquitous in all NGS labs and extremely cheap and quick. Indeed, for cluster generation on Illumina platforms cleanup is absolutely required to remove adaptor dimer and unincorporated primers, as both these species compete during cluster generation. As such, claims their method does not require this is somewhat dubious.

We thank the reviewer for pointing out this concern and we apologize that we did not make this point clear. In our understanding, a bead cleanup with $\geq 1.8X$ beads is considered "without size selection", since primer-dimers are largely kept. Therefore, we did not mean that bead cleanup is not needed. In Ampliseq's protocol, SPRI purification with 1.0X and 0.5X beads was performed (double-sided size selection). While in our protocol, only SPRI purification with 1.8X beads was performed. Therefore, we believe that adjusting the ratio of the bead to 1.8X would make a fairer

comparison between Ampliseq and our results.

As such, the fundamental claim remains whether their 3' end match function significantly advances the field. Again, the authors have done a very detailed job and should be commended, but their solution lags behind the market leader in terms of dense multiplexing (leaving aside the question of modified primers and enzymatic treatments); as well as novelty, as per the papers the authors themselves cite in their response.

While the manuscript is nicely laid out, the decision to publish would mostly be reduced down to the editors decision, as I personally believe their method is not a substantive advancement for the field.

We thank the reviewer for the comment. One of the reviewer's concerns is that Ampliseq can scale up to 20,000 primer pairs undermines the impact of SADDLE. We kindly ask the reviewer to consider an example of space probe and subway: the speed of a space probe can reach 100km/s, but it does not undermine the impact of increasing the speed of the subway from 80km/h to 300km/h. For Ampliseq and SADDLE, Ampliseq is an integrated PCR-based NGS technology, and users have limited space for further customization. On the other hand, SADDLE is a multiplexed primer design software, and the designed primer sequences could be used in any PCR application, including qPCR, nested PCR, PCR with unique molecular identifier, etc. And we believe SADDLE will still benefit researchers in the field, especially academic labs that prefer flexible and inexpensive PCR design.